# Stachydrine Hydrochloride Regulates the NOX2-ROS-Signaling Axis in Pressure-Overload-Induced Heart Failure

**DOI:** 10.3390/ijms241814369

**Published:** 2023-09-21

**Authors:** Shuang Lu, Yueyang Liang, Songru Yang, Mengwei Fu, Xiaoli Shan, Chen Zhang, Huihua Chen, Pei Zhao, Rong Lu

**Affiliations:** 1School of Traditional Chinese Medicine, SHUTCM, Shanghai 201203, China; 18132125230@shutcm.edu.cn (S.L.); 18819243869@163.com (Y.L.); yangsr777@163.com (S.Y.); fmw18810266022@163.com (M.F.); chenhuihua123456@163.com (H.C.); 2Public Experiment Platform, School of Traditional Chinese Medicine, SHUTCM, Shanghai 201203, China; shanxiaoli78@163.com; 3Department of Pathology, School of Intergrative Medicine, SHUTCM, Shanghai 201203, China; zhangchen@shutcm.edu.cn

**Keywords:** stachydrine hydrochloride, heart failure, nicotinamide adenine dinucleotide phosphate oxidase 2, reactive oxygen species, excitation-contraction coupling

## Abstract

Our previous studies revealed the protection of stachydrine hydrochloride (STA) against cardiopathological remodeling. One of the underlying mechanisms involves the calcium/calmodulin-dependent protein kinase Ⅱ (CaMKII). However, the way STA influences CaMKII needs to be further investigated. The nicotinamide adenine dinucleotide phosphate oxidase 2 (NOX2)-coupled reactive oxygen species (ROS) overproduction putatively induces the oxidative activation of CaMKII, resulting in the occurrence of pathological cardiac remodeling and dysfunction in experimental models of mice. Thus, in this study, we assessed the role of the NOX2-ROS signal axis in STA cardioprotection. The transverse aortic constriction (TAC)-induced heart failure model of mice, the phenylephrine-induced hypertrophic model of neonatal rat primary cardiomyocytes, and the H_2_O_2_-induced oxidative stress models of adult mouse primary cardiomyocytes and H9c2 cells were employed. The echocardiography and histological staining were applied to assess the cardiac effect of STA (6 mg/kg/d or 12 mg/kg/d), which was given by gavage. NOX2, ROS, and excitation-contraction (EC) coupling were detected by Western blotting, immunofluorescence, and calcium transient-contraction synchronous recordings. ROS and ROS-dependent cardiac fibrosis were alleviated in STA-treated TAC mice, demonstrating improved left ventricular ejection fraction and hypertrophy. In the heart failure model of mice and the hypertrophic model of cardiomyocytes, STA depressed NOX2 protein expression and activation, as shown by inhibited translocation of its phosphorylation, p67phox and p47phox, from the cytoplasm to the cell membrane. Furthermore, in cardiomyocytes under oxidative stress, STA suppressed NOX2-related cytosolic Ca^2+^ overload, enhanced cell contractility, and decreased Ca^2+^-dependent regulatory protein expression, including CaMKⅡ and Ryanodine receptor calcium release channels. Cardioprotection of STA against pressure overload-induced pathological cardiac remodeling correlates with the NOX2-coupled ROS signaling cascade.

## 1. Introduction

Heart failure (HF) is a complex clinical syndrome that results from either functional or structural impairment of ventricles identified by the presence of current or prior characteristic symptoms, such as breathlessness, ankle swelling, and fatigue. In the U.S., prevalent cases of HF now exceed 5.8 million, and each year, more than 550,000 new cases are diagnosed [1]. Furthermore, studies have suggested that the prevalence of HF is increasing with age [2]. Inhibitors of angiotensin-converting enzyme, mineralocorticoid receptor antagonists, beta-blockers, angiotensin receptor-neprilysin inhibitors, and sodium-glucose cotransporter-2 inhibitors are currently the core therapies for HF. Despite the significant advances in therapies and prevention, mortality and morbidity are still high, and quality of life is poor [3]. Therefore, discovering new therapies for HF is urgently required.

Traditional Chinese medicine, consisting of natural medicines and relevant extracts, has been widely accepted as an effective alternative approach for treating HF. A component of Yimucao (*Leonurus japonicas* Houtt) is stachydrine hydrochloride (STA), an active alkaloid [4]. STA can attenuate HF induced through pressure overload or isoproterenol in mice [4,5,6]. Currently, the inhibition of hyperphosphorylation of calcium/calmodulin-dependent protein kinase Ⅱ (CaMKII) is one of the anti-HF mechanisms of STA [4]. However, the way STA influences CaMKII needs to be further investigated.

A small number of reactive oxygen species (ROS) generated within cells play an important role in cell signaling. Studies have shown that overproduction of ROS contributes significantly to cardiac remodeling [7]. In patients with HF, the plasma and pericardial fluid levels of major ROS markers were increased and positively related to the severity of the condition [8]. In mice, enhanced accumulation of ROS resulted in the dilatation and remodeling of left ventricular and decreased contractile function [9], while inhibition of ROS production improved HF [10]. The ROS-generating enzyme, nicotinamide adenine dinucleotide phosphate oxidase 2 (NOX2), prevented the development of TAC-induced oxidative stress in mice [11] and was present at higher levels in the left ventricular myocardial tissue of patients with end-stage HF [12]. In addition, studies have shown that STA can inhibit ROS generation in angiotensin II-induced hypertrophic cardiomyocytes [13], which suggests that ROS potentially mediates the anti-HF effect of STA.

In the present study, we examined the role of the NOX2/ROS pathway in STA cardioprotection. Our results confirmed that STA lowered ROS generation and NOX2 protein expression in overloading pressure-induced HF in mice and the hypertrophic model of neonatal rat cardiomyocytes.

## 2. Results

### 2.1. STA Counteracts Stress-Loaded HF in Mice

To determine whether STA can protect against HF, we gavaged TAC-operated mice with low (6 mg/kg/d) and high doses (12 mg/kg/d) of STA or saline (TAC group) for 6 weeks. Low and high doses improved cardiac function and morphology compared to the TAC group. This scenario was supported by the parameters of the cardiac echocardiogram, including ejection fraction, systolic fraction, left ventricular internal diameter in diastole and left ventricular internal diameter in systole, heart weight to body weight ratio, and heart weight to tibia length ratio (Figure 1A–C). The hearts of TAC mice administrated with low and high doses of STA were significantly smaller, as shown in the physical view of the heart, heart HE longitudinal staining map, heart HE transverse staining map, and heart transverse WGA staining (Figure 1D–E). All of the above data demonstrate the alleviating effect of STA on stress-loaded HF.

### 2.2. STA Relieves the ROS and ROS-Dependent Myocardial Injury in Stress-Loaded HF in Mice

We further demonstrate whether oxidative stress is involved in the anti-HF effect of STA. Compared with the TAC group, the contents of inactive enzymes of ROS, SOD, and CAT enzyme were markedly upregulated, whereas the contents of a generating enzyme of ROS, malondialdehyde (MDA), were downregulated in the mice from the STA group (Figure 2A–C). Additionally, STA reduced cardiac fibrosis in TAC model mice, as shown by heart transverse Sirius red staining (Figure 1D). These findings indicate that STA relieves ROS and ROS-related fibrosis in pressure-overloaded-induced HF in mice.

### 2.3. STA Suppresses NOX2 Protein Expression and Activation in Stress-Loaded HF in Mice

Nicotinamide adenine dinucleotide phosphate oxidase (NOX) is a key enzymatic source of ROS in the heart [8]. NOX2, a member of the NOX family, is present at higher levels in the outer layer of human coronary arteries [9]. Thus, we confirmed whether NOX2 mediates the impact of STA on ROS. The protein expression and activity of NOX2 were increased in the TAC group and decreased after STA treatment (Figure 2D,E). NOX2 activation requires cytosolic components, p-p47phox, p67phox, and p40phox, to interact with p22phox in the cell membrane [14]. In addition, a decreased protein expression of p22phox and p-p47phox/p47phox and an increase in p67phox expression were observed in the STA group compared to the TAC group (Figure 2E,F). These results indicate that STA inhibits NOX2 expression and activity by preventing the translocation of p67phox and p-p47phox in stress-load-induced HF.

### 2.4. STA Reduces the Production of ROS and Expression of NOX2 in Cardiomyocytes Cultured with H_2_O_2_ or Phenylephrine (PE)

Next, we further confirm whether the NOX2-ROS pathway participates in STA cardioprotection in cardiomyocytes. In PE-cultured neonatal rat cardiomyocytes (NRCMs) or H_2_O_2_-cultured adult mouse cardiac myocytes (AMCMs)/rat H9c2 cardiomyocytes, STA prevented the production of ROS, as revealed by the ROS content assay kit (Figure 3A,B) and fluorescence intensity (Figure 3C). In hypertrophic NRCMs, STA downregulated the protein expression of NOX2 and p22phox in the cell membrane (Figure 4A). Immunofluorescence analysis further showed that p67phox and p47phox were ectopically expressed from the cytoplasm to the cell membrane, whereas this phenomenon was reversed to some extent after STA treatment (Figure 4B). Collectively, STA reduces ROS generation in cardiomyocytes subjected to oxidative damage, downregulates NOX2 expression, and regulates the translocation of its regulatory subunits in hypertrophic cardiomyocytes.

### 2.5. STA Sustains NOX2-Related Ca^2+^ Homeostasis and Improves Contractile Function in Cardiomyocytes under Oxidative Stress

NOX2-coupled ROS disrupts excitation-contraction (EC) coupling [15]. Therefore, we examined whether STA regulates EC coupling. In AMCMs treated with H_2_O_2_, STA decreased the contents of Ca^2+^ but enhanced the contractility (Figure 5A,B). The above results suggest that STA inhibits Ca^2+^ overload and strengthens sarcomere contractility in cardiomyocytes during oxidative stress.

### 2.6. STA Inhibits NOX2-Related Ca^2+^-Dependent Regulatory Protein Expression in Stress-Loaded HF

Since L-type Ca^2+^ channels (LTCC), calcium/calmodulin-dependent protein kinase Ⅱ (CaMKⅡ), and ryanodine receptor calcium release channel (RyR2) are involved in cardiac EC coupling [16,17,18], we observed that STA reduced the expression of phosphorylated CaMKⅡ (T286), oxidized CaMKII, and phosphorylated RyR2 (S2814) proteins in wide-type TAC mice (Figure 6A,B). In H9c2 cells incubated with H_2_O_2_, STA decreased the expression of phosphorylated CaMKII (T286), oxidized CaMKII, and phosphorylated RyR2 (S2814) proteins (Figure 7A,B). The above results suggest that STA represses the expression of CaMKⅡ and RyR2 proteins in pressure-overloaded-induced HF.

## 3. Discussion

The anti-HF effect of STA has been proven in mice subjected to TAC or isoproterenol [5,6,19]. Although the inhibition of hyper-CaMKII is one of the mechanisms of action of STA [4,5,6], upstream signaling pathways inhibiting the hyper-phosphorylation of CaMKII remain poorly understood. This study found that STA improves cardiac function and morphology and reduces ROS production and NOX2 protein expression in mice with TAC-induced HF. Moreover, STA inhibited NOX2-related calcium overload but increased sarcomere contractility while decreasing calcium-dependent regulatory protein expression in the cardiomyocytes. This evidence highlights the NOX2/ROS-regulated EC coupling as the potential pathway, shedding new light on the mechanisms of STA cardioprotection.

Previous studies showed that STA can improve HF in mice subjected to TAC or isoproterenol [5,6,19]. Herein, we recapitulated the cardio-protective effects of STA in TAC mice.

Oxidative stress results in the occurrence and progression of HF [19,20,21,22] and is positively associated with the severity of chronic HF in patients [20]. Overproduction of ROS can promote the occurrence and progression of cardiac remodeling and HF [7]. Mechanistically, the overproduction of ROS modifies RyR2 to increase its opening probability and causes oxidative activation of CaMKII, thereby leading to Ca^2+^ overload in the cytoplasm and contractile dysfunction in the heart [18]. In addition, ROS initiates the activation of a wide range of signaling kinases and transcription factors related to hypertrophy and promotes apoptosis [7]. In addition, ROS causes the proliferation of fibroblasts and activates the matrix metalloproteinases, resulting in the reconstruction of the extracellular matrix in the heart [7]. Similarly, the current study confirmed the overproduction of ROS in hypertrophic cardiomyocytes. Furthermore, increased protein expression of phosphorylated RyR2 and oxidized CaMKII, a higher content of Ca^2+^, and decreased sarcomere contractility were seen in cardiomyocytes stimulated by an oxidative stressor. After treating cardiomyocytes under oxidative stress with STA, the accumulation of ROS and subsequent EC decoupling were attenuated, suggesting a possible role of oxidative stress in the effect of STA on HF.

Inhibition of ROS generation mediates the anti-HF effect of STA. Targeting ROS and its associated enzymes has been considered a promising strategy for treating HF recently, but, at present, no drugs have been approved for clinical use [21,22]. Previous studies showed that STA depresses ROS generation in hypertrophic cardiomyocytes and breast cancer cells [13,23]. In this study, STA decreased ROS production and the protein expression of its generating enzyme, NOX2, in mice with TAC, which is indicative of the participation of NOX2 in the antioxidant ability of STA.

NOX2 is the major enzyme producing ROS in HF that is primarily found in the membrane of a cell and consists of membrane-bound gp91phox and p22phox [14]. However, activating this enzyme requires that p47phox, p40phox, p67phox, and Rac1 be involved to assemble all the enzymes [14]. Assembling and activating enzyme complexes rely heavily on the phosphorylation and translocation of Rac1 and p47phox [14]. An upregulated expression of NOX2 has been observed in the left ventricular myocardial tissue of patients with HF [10]. Our findings revealed the upregulation of NOX2 in TAC model mice and hypertrophic cardiomyocytes. Another study found that the knockout of NOX2 significantly suppresses the rise in oxidative stress in mice with pressure-overloaded-induced HF [10]. In this study, we found that STA downregulates NOX2 protein expression and suppresses the translocation of its regulatory subunits, p67phox and p47phox, from the cytoplasm to the membrane of the cells. Thus, we concluded that STA modulates NOX2 by changing the expression of NOX2 regulatory subunits.

Compared with previous studies, we discovered a correlation between NOX2-coupled ROS and STA-mediated cardioprotection. To verify the causal connection between NOX2 and STA in HF, studies using cardiac-specific NOX2 overexpressing mice are required. Furthermore, STA’s effects on oxidative stress-downstream signaling could be investigated, as a variety of downstream signal pathways within cells can be activated by oxidative stress to exacerbate myocardial pathology.

In conclusion, our findings suggest that the NOX2/ROS-regulated EC coupling might participate in the anti-HF effect of STA. These findings expand the regulatory mechanism of STA in HF and identify new targets for cardiovascular disease treatment (Figure 8).

## 4. Materials and Methods

### 4.1. Chemicals and Reagents

Stachydrine hydrochloride (STA, Cas: 4136-37-2) was acquired from the National Institutes for Food and Drug Control (Beijing, China). Phenylephrine (PE) was bought from Sigma-Aldrich (St. Louis, MO, USA). The remaining reagents were purchased as follows: H_2_O_2_ (LIRCON), NAC (N-acetyl-L-cysteine, Selleck, S1623), GSK (GSK-2795039, inhibitor of NOX2, Selleck, S1415925-18-6), LTCC (Abcam, ab253190), CaMK2 (Abcam, ab181052), P-CaMK2 (Invitrogen, PA5-37833), OX-CaMK2 (Gene Tex, GTX36254), RyR2 (Abcam, ab2868), P-RyR-2814 (Badrilla, A01031AP), P-RyR-2808 (Badrilla, 642104), NOX2 (Abcam, ab80897), p22phox (Abcam, ab75941), p67phox (Abcam, ab175293), p47phox (Affinity, AF5220), and p-p47phox (Affinity, AF3917).

### 4.2. Animals

Seven-week adult male C57BL/6J mice were obtained from Beijing Vital River Laboratory Animal Technology Co., Ltd. (Beijing, China). All the animals were housed in an environment with a temperature of 22 ± 1 °C, a relative humidity of 50 ± 1%, a light/dark cycle of 12/12 h, and given food and water ad libitum. All animal studies (including the mouse euthanasia procedure) were approved by the Animal Care and Use Committee of Shanghai University of Traditional Chinese Medicine (Shanghai, China, PZSHUTCM200703008) on 22 July 2020. Mice were randomly divided into 4 groups (8 mice per group): the sham operation group, the TAC group, and the TAC+STA groups (low and high dose groups). Mice underwent transverse aortic constriction surgery to induce pressure overload-induced HF. The sham operation group and TAC group were given normal saline by gavage once a day, while the TAC+STA groups received oral administration of STA (6 or 12 mg/kg/day). The treatment lasted for 6 weeks.

Also, male Wistar rats, 1 day old, used for neonatal rat cardiomyocyte production were provided by the Shanghai Laboratory Animal Center, Chinese Academy of Sciences, and were approved by the Animal Care and Use Committee of Shanghai University of Traditional Chinese Medicine (Shanghai, China, PZSHUTCM2301040006) on 19 December 2022.

### 4.3. Transverse Aortic Constriction (TAC) Surgery

Mice underwent transverse aortic constriction surgery to induce pressure overload-induced HF. They were anesthetized with 5% isoflurane and then placed on a heated operating table (37 °C) in a supine position with 1% isoflurane to maintain anesthesia. The chest cavity was opened to compress the aortic arch by a ligature twisted around a bent 27-gauge needle, and afterward, the needle was rapidly pulled out. The sham operation group did not have a ligature, while the other procedures were the same as in the operation group.

### 4.4. Echocardiography Analysis

Mice were anesthetized and placed supine on a heated platform (37 °C) with 1% isoflurane. Using the Small Animal Imaging System (Vevo 2100, Visual Acoustics, Toronto, Canada), the left ventricular (LV) size, ejection fraction (EF), systolic fraction (FS), left ventricular end-systolic inner diameter (LVESd), left ventricular end-diastolic inner diameter (LVEDd), end-diastolic septal thickness (IVSd), end-systolic septal thickness (IVSs), and end-diastolic posterior wall thickness (LVPWd) were evaluated.

### 4.5. Adult Mouse Cardiac Myocyte (AMCM) Isolation, Culture, and Treatment

Adult C57BL/6J mice (*n* = 3) were intraperitoneally injected with heparin (5000 IU/kg), and their hearts were removed quickly after anesthesia with pentobarbital (100 mg/kg). Then, Ca^2+^ Tyrode’s solution without Ca^2+^ Tyrode’s was infused on the Langendorff device for 5 min at 37 °C. Afterward, the heart was digested with Tyrode’s solution composed of collagenase and 50 μM Ca^2+^ for approximately 10 min. The tissue was then cut into pieces and gently blown with a large-diameter straw to collect cardiomyocytes. The concentration of Ca^2+^ in the solution reached 900 μM by the gradient compound calcium method. Rod-shaped stationary AMCMs were cultured on M199 medium in a humidified atmosphere containing 5%CO_2_/95% air at 37 °C for 1 h. The following groups were set up and co-cultured with the drug for 1 h: control group (control), model group (H_2_O_2_ 10 μM), STA group (10^−5^ M), NAC group (10 mM), and GSK-2795039 group (10^−6^ M).

### 4.6. Immunofluorescence Staining

After fixing with 4% paraformaldehyde for 15 min, AMCMs were washed with PBS. Then, the cells were incubated with immunofluorescence permeabilization buffer for 10 min and washed once with phosphate-buffered saline (PBS). Next, 5% BCA was added to the cells for incubation at room temperature for 1 h. After discarding BCA, the cells were incubated overnight at 4 °C with primary antibody (1:200) and then with Alexa Fluor (1:200) (green; CST, Danvers, MA, USA) at room temperature for 1 h in the dark. Then, WGA (red) staining (1:1000) was performed. Eventually, images were captured utilizing a Carl Zeiss LSM800 confocal microscope (Zeiss, Jena, Germany) under 488 nm and 555 nm excitation.

### 4.7. H9c2 Culture and Treatment

H9c2 cell lines were provided by the Shanghai Institute of Cell Biology (Shanghai, China). Cells were cultured in Dulbecco’s modified Eagle’s medium (DMEM, Gibco BRL, Grand Island, NY, USA) comprising 10% fetal bovine serum (FBS) in a humidified atmosphere containing 5%CO_2_/95% air at 37 °C. To set up the model of oxidative stress, a 24-hour exposure to the following conditions was performed on the cells: control group, H_2_O_2_ (250 μm), H_2_O_2_ + STA (STA 10^−5^ M), H_2_O_2_ + NAC (NAC 10^−2^ M), and H_2_O_2_ + GSK (GSK-2795039 5 × 10^−5^ M). The assessment of the influence of H_2_O_2_ on cell viability was determined using the Cell Counting Kit-8 (CCK-8).

### 4.8. Neonatal Rat Cardiomyocyte (NRCM) Culture and Treatment Protocol

The Shanghai Laboratory Animal Center, Chinese Academy of Sciences (SLACCAS), provided the male Wistar rats at 1 day of age used for NRCM production. As previously described, NRCM cultures were arranged according to a modified protocol [24]. Cells were cultured in Dulbecco’s modified Eagle medium/Nutrient Mixture F-12, supplemented with 10% FBS and 1% penicillin, in a humidified atmosphere containing 5%CO_2_/95% air at 37 °C. A 48-h exposure to the following conditions was performed on the cells: control group, PE group (100 μm), STA group (STA 10^−5^ M), and GSK group (GSK-2795039 10^−6^ M).

### 4.9. Western Blot

Extracts of proteins from cells and tissues were made using RIPA buffer comprising anti-protease and anti-phosphatase agents. The quantity of extracted protein was determined with a BCA protein quantitative assay kit (Beyotime, Shanghai, China). A total of 20 μg of lysates was loaded into SDS–PAGE gels and shifted to PVDF membranes (Merck Millipore, GER). Following one hour of sealing using 5% bovine serum albumin (BSA), membranes were incubated with primary antibody (1:1000) at 4 °C overnight. Immobilon Western Chemiluminescent HRP substrate was used to test protein bands after a 1-hour incubation with enzyme-labeled secondary antibody (1:2000). Density quantification was performed using GraphPad Prism 8.0.1 software.

### 4.10. ELISA

An appropriate amount of tissue was placed in a centrifuge tube and mixed with PBS (pH 7.2–7.4). Homogenizers were used to homogenize the sample well. A 20-min centrifugation was performed on the samples (2000–3000 rpm, 4 °C), and the supernatant was gathered. The samples and reagents were added to the Microelisa strip plate according to the steps in the instructions, and finally, an enzyme-linked immunoassay was exploited to assess the absorbance at 450 nm.

### 4.11. Detection of Intracellular ROS

The ROS content was assessed utilizing the Reactive Oxygen Species (ROS) Assay Kit (E004-1-1, Nanjing Jiancheng Bioengineering Institute, China). Cells were repeatedly washed using 0.01 M PBS, transferred to a centrifuge tube, washed twice with PBS, and centrifuged for five minutes (1000 rpm, 4 °C). The supernatant was discarded, and then PBS containing the probe was added and mixed well with a dropper. An incubation period of thirty minutes was conducted at 37 °C, and the precipitate was collected by centrifugation. In PBS, the precipitate was resuspended and assayed by enzyme-linked immunoassay.

### 4.12. DCFH-DA Staining

DCFH-DA staining was carried out as previously described [25]. AMCMs or NRCMs were stained for 10 minutes with DCFH-DA (10 μM) at 37 °C. A confocal microscope (Zeiss LSM800) was used to visualize the fluorescence.

### 4.13. Detection of Superoxide Dismutase (SOD), Catalase (CAT), and Malondialdehyde (MDA)

The SOD content was measured using the Total Superoxide Dismutase (SOD) Assay Kit (A001-3-2, Nanjing Jiancheng Bioengineering Institute, China) following the manufacturer’s instructions. The content of CAT was assessed by applying the Catalase (CAT) Assay Kit (BC0205, Solarbio, China). A proper amount of tissue was added to 1 mL of extract, ground thoroughly with a homogenizer, and centrifuged for 10 min (8000× *g*, 4 °C), and the assay was conducted using the supernatant. Lipid peroxidation (MDA) was detected utilizing a lipid peroxidation assay kit (ab118970; Abcam, Shanghai, China) following the manufacturer’s instructions.

### 4.14. Statistical Analysis

The data were presented as the mean ± standard error of the mean. The differences between two groups or among many groups were analyzed with one-way analyses of variance. *p* < 0.05 indicated statistical significance.

## Figures and Tables

**Figure 1 ijms-24-14369-f001:**
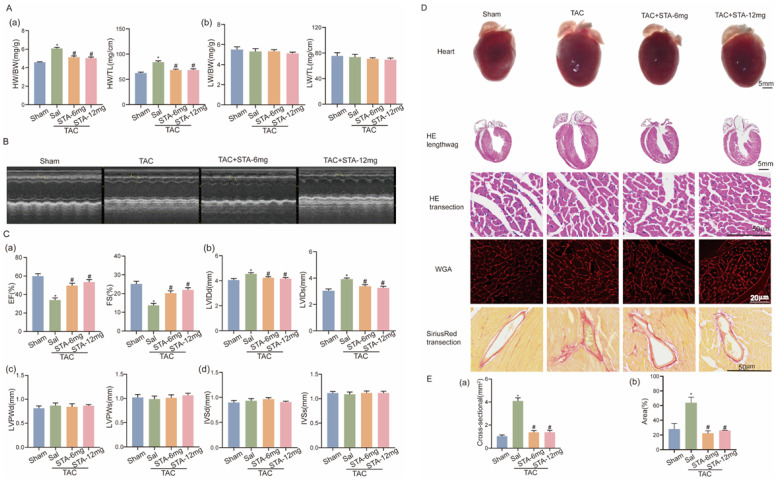
Stachydrine hydrochloride (STA) ameliorates TAC-induced heart failure in mice. (**A-a**) The heart weight to body weight and heart weight to tibia length (*n* ≥ 5 in each group); (**A-b**) the lung weight to body weight and lung weight to tibia length (*n* ≥ 5 in each group). (**B**) Representative echocardiographic images. (**C-a**) EF: left ventricle ejection fraction; FS: left ventricle fractional shortening (*n* ≥ 5 in each group); (**C-b**) end-diastolic and end-systolic internal diameters of the left ventricle (*n* ≥ 5 in each group); (**C-c**) left ventricular posterior wall thickness at end diastole and end systole (*n* ≥ 5 in each group); (**C-d**) end-diastolic and end-systolic septal thickness of the left ventricle (*n* ≥ 5 in each group), *p* < 0.05. (**D**) Heart physical images, longitudinal images of paraffin-embedded heart sections stained with HE, transverse images of paraffin-embedded hearts stained with HE, transverse images of paraffin-embedded hearts stained with WGA, and transverse images of paraffin-embedded hearts stained with Sirius red. (**E-a**) The area of cross-section of HE-stained images quantified by software for measuring cell areas in ImageJ (*n* ≥ 5 in each group); (**E-b**) the percentage of the Sirius red-stained area in images quantified by software for measuring cell areas in ImageJ (*n* ≥ 5 in each group). * vs. sham group (*n* ≥ 3 in each group), *p* < 0.05; # vs. TAC group (*n* ≥ 3 in each group), *p* < 0.05.

**Figure 2 ijms-24-14369-f002:**
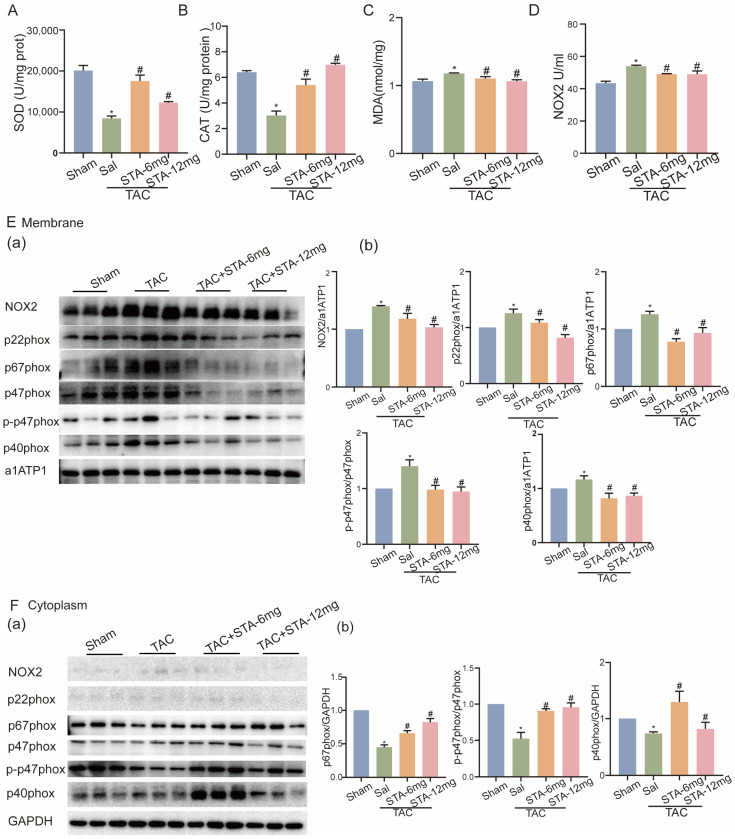
Stachydrine hydrochloride (STA) inactivates NOX2 and reduces ROS production in TAC-induced heart failure in mice. (**A**–**D**) SOD content (**A**), CAT content (**B**), MDA content (**C**), and NOX2 enzyme activity (**D**) in the H9c2 cell oxidative stress model. (**E-a**,**E-b**) Cell membrane expression of NOX2, p22phox, p67phox, p47phox, p-47phox, and p40phox. (**F-a**,**F-b**) Cytoplasmic expression of NOX2, p22phox, p67phox, p47phox, p-47phox, and p40phox. * vs. sham group (*n* ≥ 3 in each group), *p* < 0.05; # vs. TAC group (*n* ≥ 3 in each group), *p* < 0.05.

**Figure 3 ijms-24-14369-f003:**
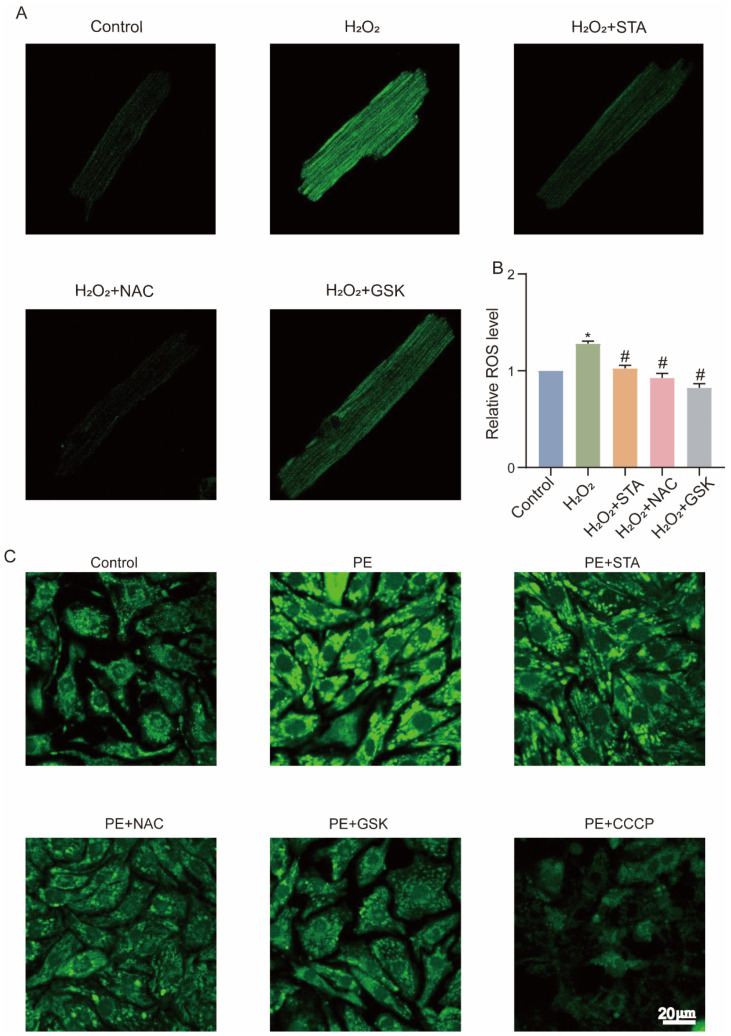
Stachydrine hydrochloride (STA) alleviates ROS production in adult mouse cardiac myocytes/rat H9c2 cardiomyocytes cultured with H_2_O_2_. (**A**) AMCMs were assessed for ROS levels by DCFH-DA staining after H_2_O_2_ and STA stimulation for 1 h. (**B**) The ROS content in the oxidative stress model of H9c2 cells. (**C**) NRCMs were assessed for ROS levels by DCFH-DA staining during PE and STA stimulation for 48 h. * vs. control group, *p* < 0.05; # vs. H_2_O_2_ group, *p* < 0.05. AMCMs: adult mouse cardiac myocytes.

**Figure 4 ijms-24-14369-f004:**
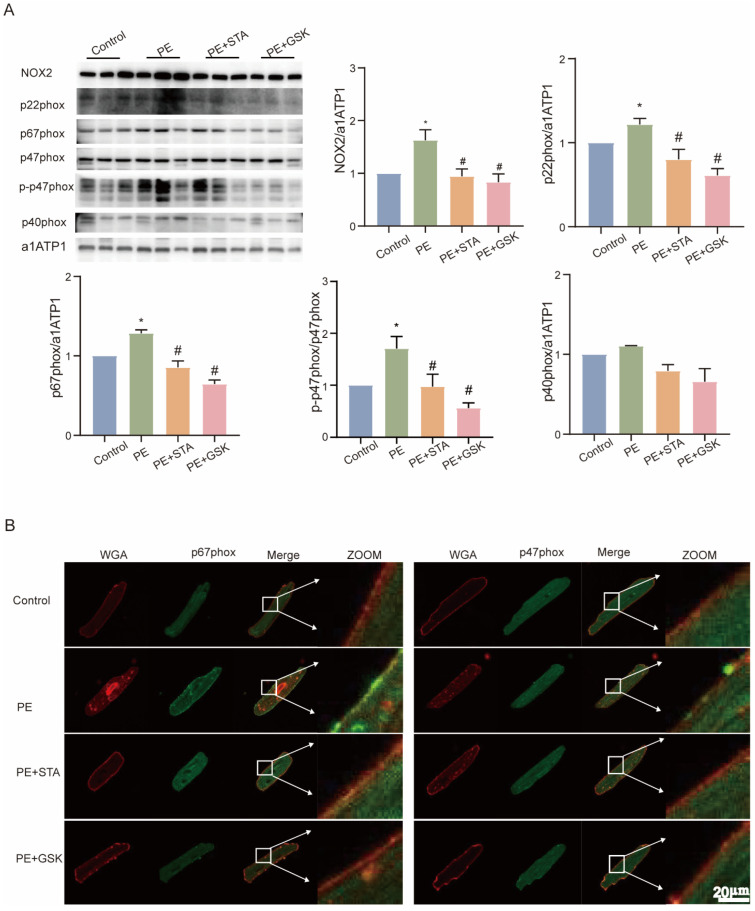
Stachydrine hydrochloride (STA) decreases NOX2 expression and membrane ectopic expression of p67 and p47 in neonatal rat cardiomyocytes cultured with phenylephrine. (**A**) Quantitative statistical plots of NOX2 and the expression of its regulatory subunits in NRCMs and grayscale values of protein bands after stimulation with PE and STA for 48 h. (**B**) Images representative of the colocalization of WGA and p67phox and p47phox; visualization of p67phox and p47phox using Alexa Fluor 488-conjugated stain (green), whereas staining of WGA using Alexa Fluor 555 (red), ×400. * vs. control group, *p* < 0.05; # vs. H_2_O_2_ group, *p* < 0.05; Wihte arrow: A local magnification view of cell membrane. NRCMs: neonatal rat cardiomyocytes; PE: phenylephrine.

**Figure 5 ijms-24-14369-f005:**
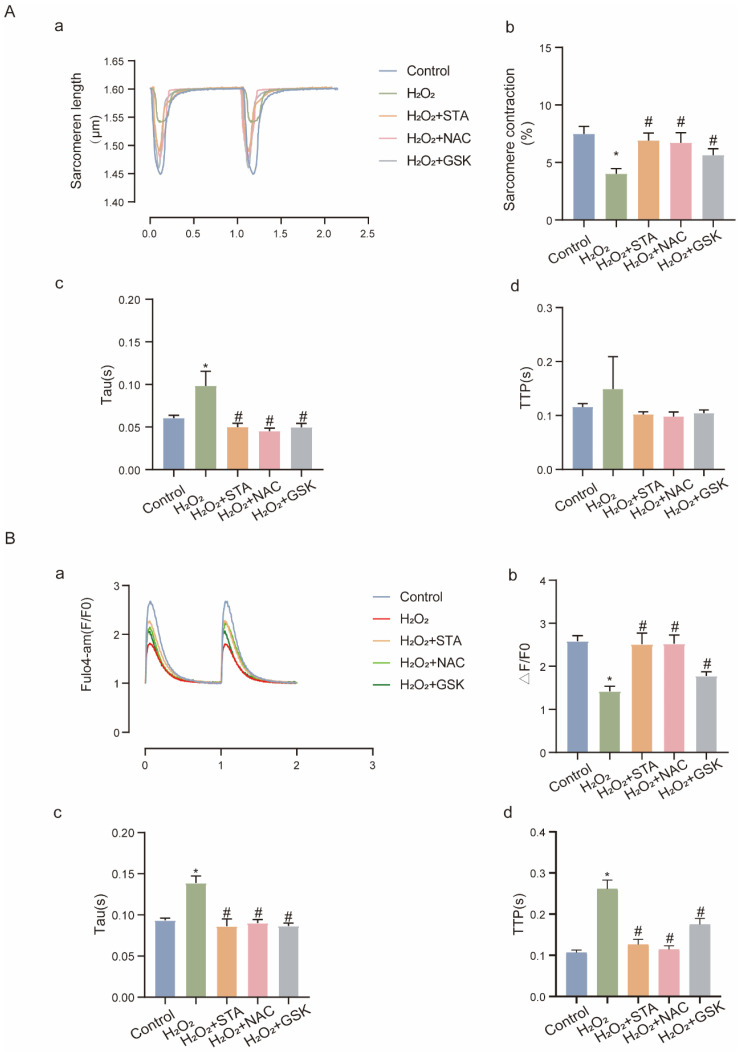
Stachydrine hydrochloride decreases calcium overload but enhances cell contractility in adult mouse cardiac myocytes cultured by H_2_O_2_. (**A-a**) Representative tracing of sarcomere contraction in AMCMs treated with H_2_O_2_ and STA for 1 h; (**A-b**–**A-d**) quantification of the sarcomere contraction, peak time, and constant of the left ventricular relaxation in each group. (**B-a**) Representative tracing of Ca^2+^ transients in AMCMs treated with H_2_O_2_ and STA for 1 h; (**B-b**–**B-d**) quantification of Ca^2+^, time to peak, and left ventricular relaxation time constant in each group. * vs. control group, *p* < 0.05; # vs. H_2_O_2_ group, *p* < 0.05. AMCMs: adult mouse cardiac myocytes.

**Figure 6 ijms-24-14369-f006:**
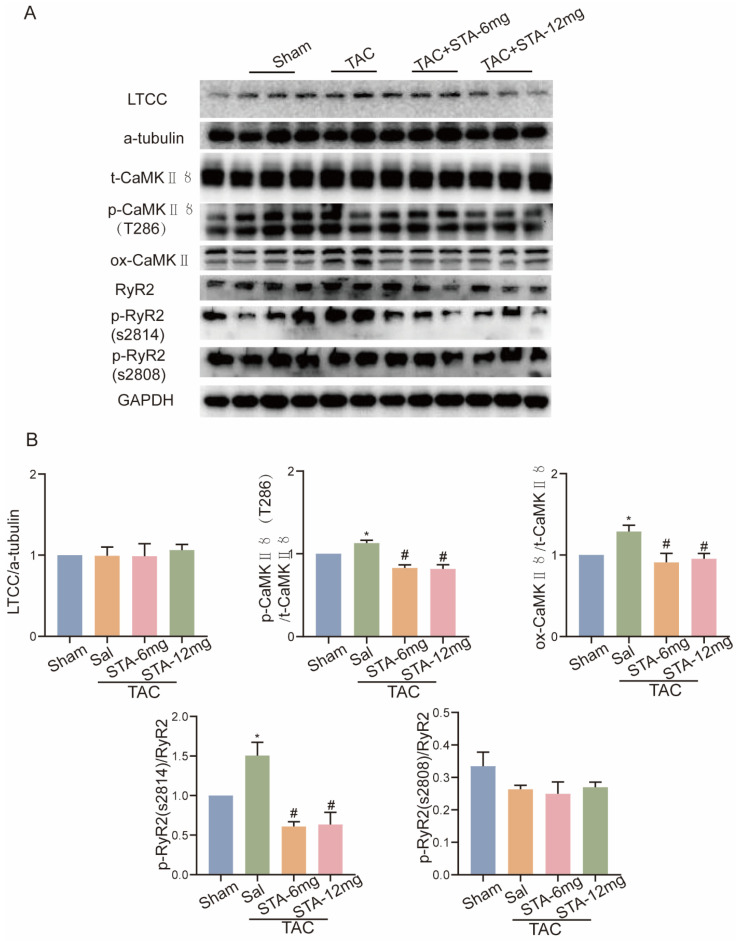
Stachydrine hydrochloride (STA) inhibits calcium-related protein expression in TAC-induced heart failure in mice. (**A**) Western blot analysis of the effect of STA on calcium-regulated protein expression in a TAC-induced heart failure mouse model. (**B**) A quantitative statistical plot of the grayscale values of the protein bands. * Compared with the sham group, *p* < 0.05; # Compared with the TAC group, *p* < 0.05.

**Figure 7 ijms-24-14369-f007:**
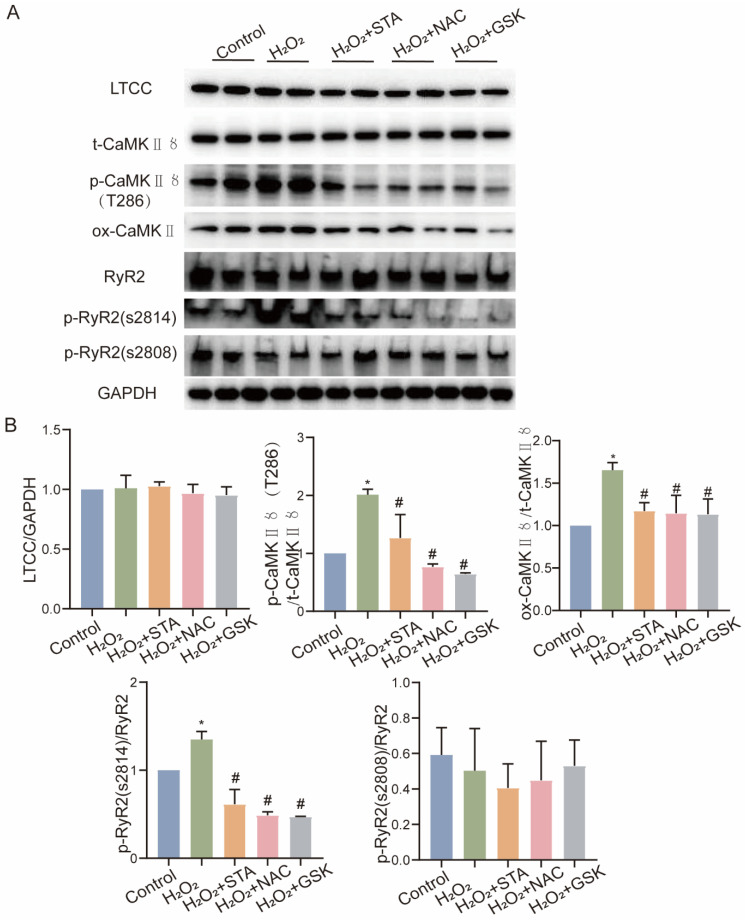
Stachydrine hydrochloride (STA) suppresses calcium-related protein expression in H9c2 cells cultured by H_2_O_2_. (**A**) A plot shows calmodulin expression in H9c2 cells treated with H_2_O_2_ and STA for 24 h. (**B**) A quantitative statistical plot of the grayscale values of the protein bands. * vs. control group, *p* < 0.05; # vs. H_2_O_2_ group, *p* < 0.05.

**Figure 8 ijms-24-14369-f008:**
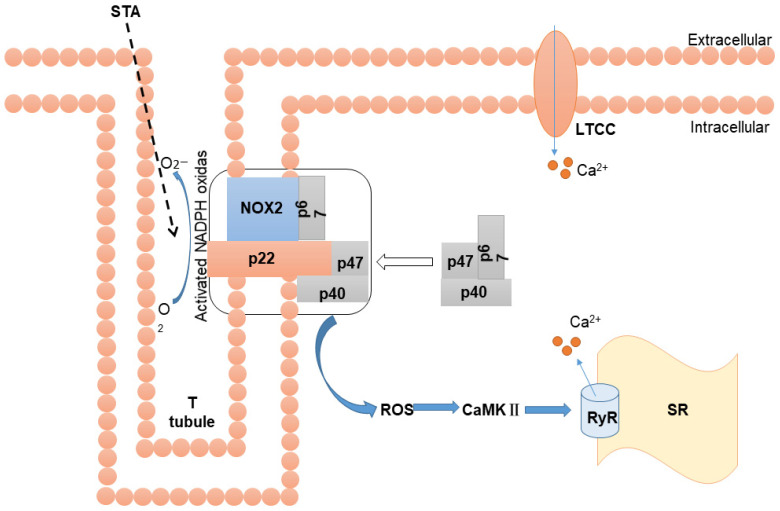
Mechanism diagram.

## Data Availability

The datasets used and/or analyzed during the current study are available from the corresponding author upon reasonable request.

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
