# Peer review of "Stachydrine Hydrochloride Regulates the NOX2-ROS-Signaling Axis in Pressure-Overload-Induced Heart Failure"

_ijms, 2023, doi:10.3390/ijms241814369_

Round 1

Reviewer 1 Report

In the present article "Stachydrine Hydrochloride regulates the NOX2-ROS-Signaling Axis in Pressure-Overload-Induced Heart Failure" the authors investigate the effects of Stachydrine Hydrochloride on heart failure mice.

Some comments:

- In Introduction is cited the 2017 guideline of Heart failure. We have newer guidelines, for example the European guideline for heart failure from 2021 updated this year, were the strategy for treatment is a little different. Please update.

- the abbreviation for Stachydrine Hydrochloride - STA appears in the text sometimes Sta. Please standardize the content.

- in subchapter 2.6 the line “Next, cells were mixed with BCA for 1 and incubated with primary antibody” is unclear please clarify.

Author Response

Thank you very much for taking the time to review this manuscript. Your suggestions indeed improve the quality of my article. Please find the detailed responses below and the corrections highlighted in the re-submitted files.

Comments 1: In Introduction is cited the 2017 guideline of Heart failure. We have newer guidelines, for example the European guideline for heart failure from 2021 updated this year, were the strategy for treatment is a little different. Please update.

Response 1: Thank you for pointing this out. I agree with this comment. Therefore, I have updated the strategy for the treatment of heart failure (page 2, paragraph 1, lines 4-8).

Comments 2: the abbreviation for Stachydrine Hydrochloride - STA appears in the text sometimes Sta. Please standardize the content.

Response 2: Agree. I have standardized the abbreviation for Stachydrine Hydrochloride as STA (page2, paragraph 4, and line 2; page 3, paragraph 5, and line 11; page 4, paragraph 1, and line 6; page 4, paragraph 2, and line 7; page 5, paragraph3, and line 11).

Comments 3: In subchapter 2.6 the line “Next, cells were mixed with BCA for 1 and incubated with primary antibody” is unclear please clarify.

Response 3: Agree. I have completed the content of the line in subchapter 2.6 (page 3, paragraph 6, and lines 3-6).

Reviewer 2 Report

The authors investigates the cardioprotective effects of Stachydrine Hydrochloride (STA) in pressure-overload-induced heart failure models in mice and in vitro cell cultures. The authors focus on the NOX2-ROS signaling axis and its role in cardiac remodeling and dysfunction. The study employs a variety of experimental models, including transverse aortic constriction (TAC)-induced heart failure in mice, phenylephrine-induced hypertrophic models in neonatal rat primary cardiomyocytes, and H2O2-induced oxidative stress models in adult mouse primary cardiomyocytes and H9c2 cells. The paper concludes that STA alleviates ROS and ROS-dependent cardiac fibrosis, suppresses NOX2 protein expression and activation, and enhances cell contractility. 

It is a well done study and has been presented clearly. I have the following minor suggestion:

The introduction provides a good background on heart failure and the need for alternative therapies. However, it could be improved by:

Providing a clearer statement of the research question.

Offering more context on why the NOX2-ROS signaling axis is important in heart failure.

Author Response

Thank you very much for taking the time to review this manuscript. Your suggestions indeed improve the quality of my article. Please find the detailed responses below and the corrections highlighted in the re-submitted files.

Comments 1: Providing a clearer statement of the research question.

Response 1: Thank you for pointing this out. I agree with this comment. Therefore, I have refined the statement of the research question (page 2, paragraph 2, lines 2-6).

Comments 2: Offering more context on why the NOX2-ROS signaling axis is important in heart failure.

Response 2: Agree. Since the role of ROS in heart failure has been discussed in the text, I added information about the influence of NOX2 on ROS in heart failure (page2, paragraph 3, and lines 7-10).